# Deep Learning of Intrinsically Motivated Options in the Arcade Learning Environment

## Abstract

Although Intrinsic Motivation allows a Reinforcement Learning agent to generate *directed* behaviors in an environment, even with sparse or noisy rewards, combining intrinsic and extrinsic rewards is non trivial. As an alternative to the widespread method of a weighted sum of rewards, Explore Options let the agent call an intrinsically motivated agent in order to observe and learn from interesting behaviors in the environment. Such options have only been established for simple tabular cases, and are unfit to high dimensional spaces. In this paper, we propose Deep Explore Options, revising Explore Options within the Deep Reinforcement Learning paradigm to tackle complex visual problems. Deep Explore Options can naturally learn from several unrelated intrinsic rewards, ignore harmful intrinsic rewards, learn to balance exploration, but also isolate exploitative or exploratory behaviors. In order to achieve this, we first introduce J-PER, a new transition-selection algorithm based on the interest of multiple agents. Next, we propose to consider intrinsic reward learning as an auxiliary task, with a resulting architecture achieving $50\%$ faster wall-clock speed and building a stronger, shared representation. We test Deep Explore Options on hard and easy exploration games of the Atari Suite, following a benchmarking study to ensure fairness. Our results show that not only can they learn from multiple intrinsic rewards, they are a very strong alternative to a weighted sum of rewards, convincingly beating the baselines in 4 of the 6 tested environments, and with comparable performances in the other 2.

## 1 Introduction

In Reinforcement Learning (RL), an agent is sequentially given states and needs to perform actions in order to maximize obtained extrinsic rewards $r_e$. The agent is therefore deeply tied to the reward signal, and tends to fail when said signal is sparse or noisy. When the environment is very complex or high-dimensional, it is desirable for the agent to explore in a *directed* way (Thrun, 1992), i.e. explicitly looking for new knowledge and experiences. One of the most common ways to generate such task-independent, directed behaviors is through *intrinsic motivation* (IM), i.e. an alternative reward signal $r_i$ to spur curiosity and entice behavior exploration (Oudeyer & Kaplan, 2009; Schmidhuber, 2010). IM biologically refers to the natural tendency of organisms to explore.

One of the most common benchmarks for Deep RL agents has been the Arcade Learning Environment (ALE, Bellemare et al. (2013)), consisting of Atari video-games. In order to solve the most challenging, so-called *hard-exploration* games of the domain (Bellemare et al. (2016)), state-of-the-art Deep RL methods have integrated IM in complex learning mechanisms, and finally managed to overcome human-level play in all 57 games (NGU and Agent57, Badia et al. (2020b;a)). However, a benchmarking study of IM techniques (`benchmark`, Taiga et al. (2019)) shows that standalone IM methods still struggle to perform greatly over all games, i.e. both hard- and easy-exploration games. It is shown that, when a same $\beta$ hyper-parameter is chosen for all games in the usual expression $r_t = r_e^t + \beta r_i^t$, there is no improvement over the more traditional methods of exploration.

This wide-spread weighted-sum (`WS`) scheme $r_t = r_e^t + \beta r_i^t$ has been discussed to come with several drawbacks, such as wasting world knowledge and transfer potential, hard fine-tuning of the $\beta$ hyperparameter, and bad scaling to multiple IM functions. In a tabular setting, Explore Options (`EO`, Bagot et al. (2020)) have been proposed as an alternative to the `WS` to answer these concerns. The agent is divided into the Exploiter, trained exclusively with the extrinsic reward $r_e$, and the Ex-

plorer, trained exclusively with the intrinsic reward $r_i$. The Exploiter can call the Explorer through an *option* (Sutton et al., 1999), i.e. an additional action, to explore for a fixed amount of time. This decoupling prevents the forgetting of the exploratory behavior, maps naturally to multiple rewards, and theoretically lets the agent explicitly learn when to explore. However, because of their tabular nature, these options are inadequate for function approximation, but it is a crucial element to allow the agent to generalize the option over states and effectively learn to balance exploration. In addition, parameter sharing is necessary to prevent an overhead scaling linearly in the number of agents. We revise Explore Options into Deep Explore Options, a new method for combining reward signals in Deep RL. We introduce several key changes to general IM approaches, and showcase their performance in the ALE. To provide fair and controlled comparisons, we match the algorithm and hyperparameters used in the `benchmark`. The contribution of our work is fourfold:

- We revise Explore Options within Deep RL to propose Deep Explore Options as a strong alternative to a weighted sum when using intrinsic rewards, extending the `benchmark`;
- We empirically show several properties of DeepEOs: learning from more than one IM reward at once, selecting which to use or ignore, and extracting the exploiting behavior;
- We propose auxiliary task learning of the intrinsically motivated agents, inducing a $50\%$ faster runtime for a similar performance and stronger representation;
- We introduce J-PER as an extension of Prioritized Experience Replay (Schaul et al., 2016) for multi-agent buffer sampling.

In Section 2, we provide background to build Deep Explore Options. Next, we introduce our method and discuss the introduced elements in Section 3. In Section 4, we provide experiments in Atari following the `benchmark`. We go over existing work in the field and discuss desirable properties of methods to combine reward signals in Section 5.

## 2 BACKGROUND AND EXPLORE OPTIONS

### 2.1 REINFORCEMENT LEARNING, OPTIONS, MOTIVATION

We use the standard RL setting (Sutton & Barto (2018)), modelling the environment as a Markov Decision Process $(\mathcal{S}, \mathcal{A}, \mathcal{R}, p, \gamma)$ where $\mathcal{S}$ is the set of states, $\mathcal{A}$ is the set of actions, $\mathcal{R} \subset \mathbb{R}$ is the set of rewards, $p : \mathcal{S}, \mathcal{A}, \mathcal{S}, \mathcal{R} \rightarrow [0, 1]$ is the dynamics function, and $\gamma$ is the discount factor. The goal of RL is to maximize the expected sum of discounted rewards from any starting state.
Options (Sutton et al. (1999)) refer to temporally extended actions. An option is defined as a triple $(\mathcal{I}, \pi, \beta)$, where $\mathcal{I} \subset \mathcal{S}$ is the option's initiation set, i.e. states in which the option can initiate; $\pi$ is the option's policy; and $\beta : \mathcal{S} \rightarrow [0, 1]$ is the option's termination condition.
We assume one or several intrinsic reward functions $f_{ir}(s, a, s') = r_i$ to generate a reward that we are interested in learning from. These can attempt to motivate directed exploration behaviors, but also any other behavior in the environment. An overview of the literature populating the field can be found in Section 5.2.

### 2.2 EXPLORE OPTIONS

Explore Options (`EO`, Bagot et al. (2020)) have been introduced as an alternative to a weighted sum of rewards, in order to ease fine-tuning and prevent loss of the exploratory behavior. The method consists in decoupling the Agent into an Explorer, trained with the intrinsic reward $r_i$, and an Exploiter, trained with extrinsic reward $r_e$. Switching from Exploiter to Explorer is done through the Explore Option, which the Exploiter can use at any time to let the Explorer act for a fixed amount of steps $c_{switch}$. The main intuition is that the Exploiter calls the Explorer to execute directed exploration, while the Explorer teaches the Exploiter how to explore and reach new rewards through off-policy learning. Explore Options have only been introduced in a tabular setting, where they can most easily be checked for stability and robustness. However, by design they only make sense in a function approximation setting: the option learning requires generalization in order to call the option in states where little is known, and therefore directed exploration is required. In addition, `EO`s enable learning from multiple rewards, but we cannot afford to scale training time linearly in the number of intrinsic reward functions available. Some form of parameter sharing will be required to jump to tens of Explorers.

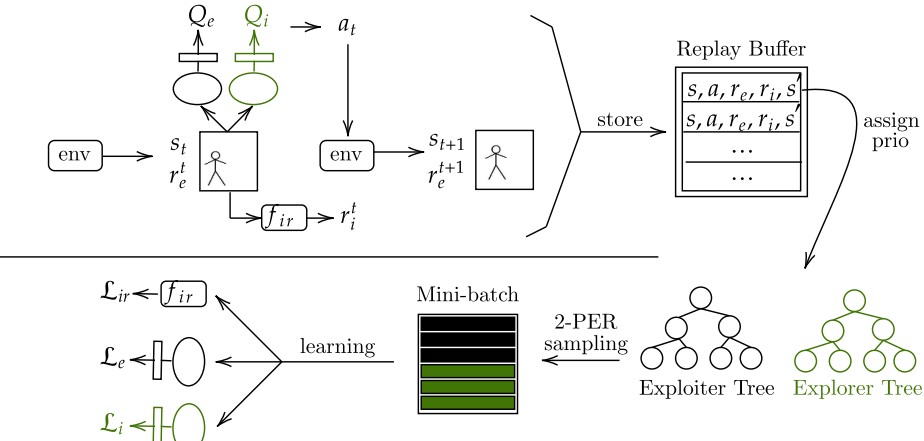

Figure 1: Deep Explore Option framework. During interactions, the state is passed through both the Explorer and Exploiter Networks and one of them acts according to the Explore Option. Transitions with intrinsic reward are stored in the replay buffer, the corresponding `PER` priorities are assigned by both Exploiter and Explorer according to their respective loss. The mini-batch is then created using transitions according to each agent's priorities following 2-`PER` sampling of both trees, and used for training of all components.

## 3   DEEP EXPLORE OPTIONS AND RAINBOW

Fig. 1 gives an overview of the general Deep Explore Option framework, of which we go into more detail in the following subsections. Implementation details of minor importance are reported in Appendix A.1.

### 3.1   ALGORITHMIC BACKBONE & TRAINING

**Benchmarking Study & Random Network Distillation**   Experimental protocols in IM tend to be chaotic, with new approaches usually introduced on arbitrary environments and algorithms, and only compared to a non-IM baseline. Taiga et al. (2019) (`benchmark`) provides an experimental setup to fairly compare IM methods in the ALE, using the `WS` scheme. It uses the Dopamine framework (Castro et al., 2018), which provides an implementation of the Rainbow agent (Hessel et al., 2018) with its 3 majors improvements on DQN (Mnih et al., 2015): Prioritized Experience Replay (Schaul et al., 2016), n-step returns, and Categorical DQN (C51, Bellemare et al. (2017)). We match this setting and all hyperparameters using the DeepRL PyTorch implementation (Zhang, 2018). The `benchmark` compares several intrinsic reward functions; we focus on Random Network Distillation (RND, Burda et al. (2019b)), as it is the simplest, most used method, and is currently part of state-of-the-art methods (Badia et al., 2020b;a). RND uses a fixed, randomly initialized *target* network $f : \mathcal{S} \to \mathbb{R}^m$ and a learnt *predictor* network $\hat{f} : \mathcal{S} \to \mathbb{R}^m$. The method generates an intrinsic reward $r_i = f_{ir}(s, a, s') = \left\|\left|\hat{f}(s') - f(s')\right|\right\|_2^2$ by distilling the target into the predictor, using the loss as motivation signal. This will therefore lead the agent towards less-visited states, that still show a high distillation error. In order to keep our return estimates within C51's usual $[-10, 10]$ range, we scale the RND reward by $\eta = 0.01$. This is not to be confused with the $\beta$ from `WS`, as this $\eta$ is vastly easier to tune: it is only meant to keep the RND returns within a reasonable scale and never meant to balance different reward signals.

**Training protocol and Off-Policy Learning**   We build two separate Rainbow agents, using architectures and hyperparameters from the `benchmark`, as our Exploiter and Explorer. We add an action to the Exploiter, corresponding to the `EO`. All observed transitions are added to the same buffer, regardless of the agent currently interacting. We train both agents simultaneously, off-policy,

using the same sampled buffer data, masking the option transitions for the Explorer. The Explorer also is not provided the `done` signal, since it might leak information about the task, rush agents with negative rewards, and interfere with exploration (Burda et al., 2019a).

## 3.2 `J-PER`: MULTI-AGENT PRIORITIZED EXPERIENCE REPLAY

Rainbow uses Prioritized Experience Replay (`PER`, Schaul et al. (2016)) to sample more interesting transitions from the buffer, where the notion of priority of a transition is its associated Rainbow loss. Transition priorities are stored in a Sum Tree to enable efficient sampling. Since we now have two agents, and therefore two notions of priority, we need to adapt the `PER` algorithm. One possible method is to duplicate the `PER` algorithm and pass each agent its own interesting data independently, but this would scale with the number of Explorers in the case of multiple intrinsic reward functions.

Instead, we infer and find that *the Explorer and Exploiter might benefit from observing each other's interesting data*. Indeed, `PER` assigns a priority based on the agent's loss, but the transition needs to be observed before the loss is updated – so a transition that was of little interest up to a certain point can later become relevant because it stood in a rewarding trajectory. Such transitions will not be particularly prioritized by `PER`. On the other hand, the Explorer is bound to select transitions that are visually appealing, or that lead to visually appealing states. In visual environments like video-games, such transitions have a strong chance to be relevant regardless of the reward we optimize, as they often correlate with key transitions of the MDP. We execute this idea by keeping two `PER` priority trees, one for each agent. For a batch size of $b_s$, we select a split weight $w_s \in [0, 1]$ and sample $b_s w_s$ transitions according to the Exploiter tree, and $b_s (1 - w_s)$ transitions according to the Explorer tree. This means that both agents will learn off-policy from the same data, and see transitions that the other considers interesting. We refer to this method for 2 agents as 2-`PER` (`J-PER` in general). Importance Sampling is simply done using the total probability of observing a transition $\tau_j$, $\mathbb{P}(j) = w_s \mathbb{P}_e(j) + (1 - w_s) \mathbb{P}_i(j)$ where $\mathbb{P}_e(j)$ and $\mathbb{P}_i(j)$ correspond to the probability for $\tau_j$ to be sampled according to the Explorer's and Exploiter's priorities respectively.

## 3.3 AUXILIARY TASKS

Sharing parameters of the model with auxiliary tasks in RL has been shown to provide a denser learning signal and accelerate learning by building a strong environment-aware representation (Jaderberg et al., 2017), but the learnt prediction and control models are rarely used beyond enriching the loss.

We propose to consider our Explorer learning, i.e. the learning of a fully intrinsically motivated policy or value function, as an auxiliary task. Rather than keeping separate vision modules (Convolutional Layers), we share the representation of both agents and only separate the control modules (Fully Connected Layers). The shared visual representation therefore needs to be general enough to be relevant for both our Explorer and Exploiter. Each head and independent parts of the same optimizer can adjust for their own reward scale. In our experiments, this architecture has about $50\%$ faster wall-clock speed than separate networks, as less parameters are used. It is important to note that a goal-oriented architecture such as UVFA (Schaul et al. (2015)) would not allow for efficient transfer learning in our case, because it is not trivial to reset the Exploiter behavior while conserving the Explorer behavior. Instead, our multi-headed architecture allows for efficient swapping, isolation or resetting of control heads, meaning that adding an Explorer or switching tasks can be done while preserving both the Explorer behaviors and general visual representation.

We combine the losses through $\mathcal{L} = \mathcal{L}_e + \lambda \mathcal{L}_i$ with $\lambda = 1$. This, again, is not to be confused with the $\beta$ from `WS`, since each head adjusts for their own reward scales, meaning that $\lambda$ does not carry the duty of reward balancing. In addition, $\lambda$ required no tuning in our experiments, partly because the KL losses of C51 often have similar scales, so we do not risk one gradient overshadowing another.

## 3.4 MULTIPLE EXPLORERS

**Learning from several intrinsic rewards** Intrinsic reward functions $f_{ir}$ are often based on heuristics and intuition regarding what signal would motivate the agent to produce the most interesting and beneficial behaviors. The range of possible behaviors produced by Intrinsic Motivation is extremely wide and widens with the complexity of the environment – in a Visual Navigation setting for example, we can motivate going fast, grabbing or observing objects, covering ground, changing rooms,

opening doors... Therefore, it becomes relevant to learn from *multiple* (say, $J$) intrinsic rewards at once, since even exploration-driven rewards might complement each other in different settings and environments (Matusch et al., 2020). The `WS` approach very hardly scales with several intrinsic reward functions $f_{ir,j}$, since it would lead to the weighed sum $R = R_e + \sum_j \beta_j R_{i,j}$, where the tuning of all $\beta_j$ and the cacophony of all reward signals would make finding a stable agent virtually impossible. We can simply visualize that an agent with $r_{speed} + r_{grab\_objects}$ would make for a very distraught joint behavior. At the other end, `EO`s scale naturally with several $f_{ir,j}$, by simply adding options and associated Explorers per $j$. The only practical issue will arise in edge cases where $J$ dominates the action space, hence primitive action learning might be impeded, or if the scarce relative interacting of all Agents begets the necessity of Offline RL (Fujimoto et al., 2019; Levine et al., 2020)).

**`ConstNeg` intrinsic reward**  Because the amount of possibly interesting intrinsic rewards is so vast, some are bound to be sub-optimal or even harmful to our agent, which will therefore need to be able to ignore them and focus on the most relevant signals. To demonstrate this, we introduce `ConstNeg`, a constant negative intrinsic reward $f_{ir} = -0.025$ that behaves like a living reward. Crucially, we do *not* provide the `done` signal to indicate life loss. For an agent that cannot differentiate between termination signals between the reward functions, like a naive `WS`, `ConstNeg` would simply hurt performance by rushing the agent towards life loss[1]. Instead, without the termination signal, the agent will be pushed off the starting point and have to explore to make sure that all regions of the state space are harmful. While this can originally help exploration, the agent will quickly learn that no behavior leads to reward, eventually rendering the introduced behavior useless. In other words, this naive and trivial-to-implement function induces a behavior that we might want to learn from, but not heavily rely upon. With it, we mean to show that *(i)* a single intrinsic reward function does not need to generate *all* interesting behaviors, so long as we can combine it with other intrinsic reward signals; therefore *(ii)* a method to combine reward signals needs to be able to scale to several signals, and be robust to harmful signals. Our experiments with this intrinsic reward will show that Deep Explore Options allows for both.

# 4 EXPERIMENTS

## 4.1 EXPERIMENTAL PROTOCOL

We evaluate Deep Explore Options with RND with the same core algorithm, architecture and hyperparameters as the Rainbow implementation from the `benchmark`. Architectural details can be found in Appendix A.2. We compare our method to two baselines provided in the `benchmark`: Rainbow with $\epsilon$-greedy ("`rainbow`") and Rainbow with an RND intrinsic reward using `WS` ("`rainbow+rnd`"). We refer to our method as "DeepEO". Agents are trained over 5 seeds for up to 100 million frames on 6 of the most relevant games of the `benchmark`– games in which methods performed most differently, indicating difficulty and importance of the IM approach. We use 3 hard-exploration games: MONTEZUMA'S REVENGE, GRAVITAR, PRIVATEEYE; and 3 easy-exploration games: SPACE INVADERS, ASTERIX and SEAQUEST. Following the `benchmark` and recommended practices (Machado et al., 2018), we use $\varsigma = 0.25$ sticky-actions and no termination on life loss. The hyperparameters we introduced take values $c_{switch} = 15$ and $w_s = 0.1$ unless stated otherwise, and are studied in subsection 4.3

## 4.2 RESULTS – ARCHITECTURES

We start by comparing our 3 main `DeepEO` architectures. First, the `Separate` Exploiter and (RND-based) Explorer networks; then the `Shared` Exploiter-Explorer visual representation (section 3.3); and finally the `Multiple` Explorers, i.e. branching to our `Shared` representation an Explorer trained with the potentially harmful `ConstNeg` intrinsic reward (section 3.4). We carry this study out on GRAVITAR and MONTEZUMA, as they provide the most disparity in performances of the baselines. We then perform additional experiments on the 4 other games with `Multi` only, as our main contribution. We compare `DeepEO Multi` to the two baselines from the `benchmark`, `rainbow` and `rainbow+rnd`.

---

[1]In GRAVITAR and MONTEZUMA, the optimal weighted-sum policy with `ConstNeg` is to die quickly.

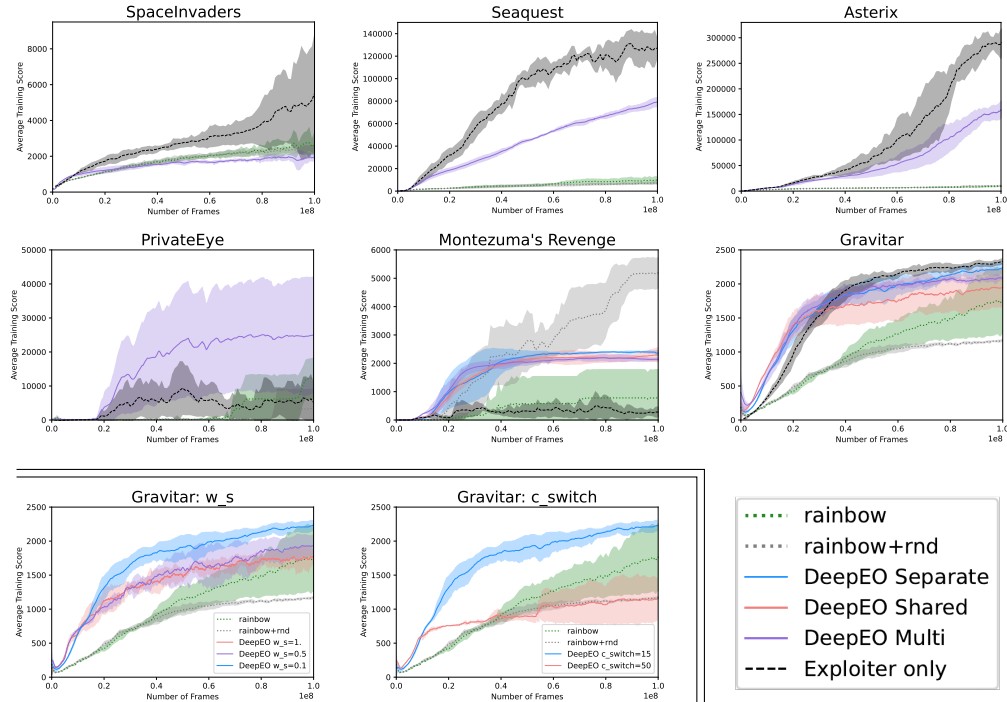

Figure 2: Deep Explore Option architectures against Rainbow baselines in Atari. Shaded areas represent $\pm$ standard deviation over 5 runs (3 for hyperparameter studies). Top row: easy-exploration games; middle row: hard-exploration games. Bottom row, left: hyperparameter studies for $w_s$ and $c_{switch}$ in Gravitar.

**Architectures Comparison**   As can be observed in Fig 2, all `DeepEO` architectures outperform both baselines in GRAVITAR, while achieving good performances in MONTEZUMA, with the better overall performances.

The `Shared` and `Multi` appear slightly less performant but very comparable to `Separate`. However again, this architecture is $50\%$ faster for 2 agents, about $100\%$ faster for 3 Agents, and builds a stronger representation – we believe the choice should be obvious, especially as we learn from more intrinsic reward functions. Crucially, `Multi` was able to learn consistently even with the addition of the naive `ConstNeg` both to its behavior (as an Explorer) and visual representation (in the shared architecture). We believe this is a major step from the Weighted Sum, which wouldn't be able to adjust to this reward. It is interesting to notice that `Multi` even seems slightly more robust than `Shared`; as if this additional Explorer could act as some form of regularization. This will be investigated in future work.

**Baslines Comparison**   We now widen our focus to the 6-game results of `Multi` versus `rainbow` and `rainbow+rnd`. We additionally report Exploiter-only performance, i.e. isolating the Exploiter to only access primitive actions, in order to extract the learnt reward-maximizing behavior. This is obtained in an evaluation fashion, every million steps over 5 episodes, with the current $\epsilon$.

We observe that the Deep Explore Option tremendously improves on the weighted sum, with very clear domination on most games. The method is also consistently faster in early training, reliably finding sources of reward faster than the baselines. Very interestingly, the Exploiter-only testing performance vastly improves the results on easy-exploration games, but is much worse than the training performance in hard-exploration games; with GRAVITAR as an outlier. This aligns with intuition, as hard-exploration games are the ones where we most expect the options to be important, hence stripping the agent off of its Explorers there should have the most impact.

In order to provide more intuition into the Explore Option and Explorers' workings, we provide Q-value estimates and interaction frequencies in Appendix B.

### 4.3 RESULTS – HYPERPARAMETERS

$w_s$ **hyperparameter**  $w_s$ controls the 2-PER ratio of transitions interesting to the Exploiter or Explorers that are passed for off-policy learning. $w_s = 1$ means passing only transitions selected by the Exploiter, i.e. for the task reward, while $w_s = 0$ would mean passing transitions only relevant for the Explorer – intuitively, where new data can be observed. This hyperparameter turns out to be of importance in the performance, but required little tuning. We tested it over 3 seeds and the values $w_s \in \{0.1, 0.5, 1\}$ on Gravitar and report the impact on performance on the Separate architecture. Note that we do not study $w_s = 0$ since this would prevent the Exploiter from learning the option, as the Explorer ignores it.

Hyperparameter studies are performed over 3 seeds on GRAVITAR and can be seen in the bottom row of Fig 2; left for $w_s$. We observe that the agent does not benefit from observing a lot of exploiting data ($w_s \approx 1$), but rather benefits from observing data selected by the Explorer. We infer that this is because Explorers are generally attracted to visually appealing transitions, which are often very relevant in video-games, while the Exploiter's priority for this transition yet has to be updated to reveal interest.

$c_{switch}$ **hyperparameter**  In Bagot et al. (2020), the authors linked EO's $c_{switch}$ hyperparameter to WS' $\beta$, since both dictate the intensity of exploration in their respective approach. It was shown that $c_{switch}$ was much more robust and lead to overall better performances than any value of $\beta$ in a simple setting. While we do not have access to the benchmark's per-$\beta$ performance, we replicate this study in Atari by comparing two different values of $c_{switch}$. It is expressed in number of steps the Explorer acts, so we study $c_{switch} \in \{15, 50\}$.

Results can be seen on the bottom row of Fig 2, middle. We observe that a higher value of $c_{switch}$ seems to severely harm performance; we infer that this is due to higher option-value variance that makes learning the option less stable. Indeed, unlike a tabular setting, in a complex visual setting and with function approximation, the option is harder to predict and the agent will attempt to generalize. We advocate for smaller values, as they are easier to learn; in addition, they can cover all behaviors generated by the higher values, if the Exploiter learns to spam the option when it is relevant.

## 5  RELATED WORK & DISCUSSION

### 5.1  EXPLORATION WITHOUT INTRINSIC MOTIVATION

Methods that do not incorporate Intrinsic Motivation were state-of-the-art on Atari until recently. Such methods generally injected noise in the action or parameter space to explore. In Value-based methods, this concerns mainly $\epsilon$-greedy (Mnih et al., 2015), which randomly explores with some probability at each step; and Noisy Nets (Fortunato et al., 2018; Hessel et al., 2018), which introduce a learnt noise in the parameters. In Policy-based methods, entropy maximization (Haarnoja et al., 2018) takes advantage of the stochastic policy formulation to motivate noisier policies.

### 5.2  INTRINSIC MOTIVATION

IM (Aubret et al., 2019) is generally used for *knowledge acquisition* about the environment, but can also encompass *skill learning* (Eysenbach et al., 2019), where the agent learns a set of temporally-extended actions to help solve the task. In knowledge acquisition, motivating exploration is by far the most common role of IM, and is often divided into *state novelty* – which encompasses count-based methods (Bellemare et al., 2016; Ostrovski et al., 2017; Burda et al., 2019b) –, *prediction error* (Pathak et al., 2017) and *information gain* (Schmidhuber, 2010). A notable recent method is Never Give Up (NGU, Badia et al. (2020b)), combining intra-episodic and inter-episodic intrinsic rewards to motivate thorough exploration. The $\beta$ fine-tuning is done by a spectrum of actors, taking advantage of the UVFA goal-oriented framework to pass $\beta$ as input (Schaul et al., 2015). Agent57 (Badia et al., 2020a) improves on NGU by focusing the $\beta$ fine-tuning on the most relevant values using a Multi-Armed Bandit setting, and separating the value functions to provide stability.

| Method Name | Value Functions | IM usage | Exploratory behavior | Transfer | Multiple IM signals | Auxiliary Task |
|---|---|---|---|---|---|---|
| `benchmark` | Combined | WS | Merged | No | No | No |
| RND | **Separate** | WS | Merged | ? | No | **Yes** |
| Explore&Exploit | **Separate** | **Decoupled** | Random switch | **Yes** | **Yes** | No |
| NGU | Combined | WS | Merged | No | No | **Yes** |
| Agent57 | **Separate** | WS | Merged | **Yes** | No | No |
| **DeepEO** | **Separate** | **Decoupled** | **Option call** | **Yes** | **Yes** | **Yes** |

Table 1: Comparison of methods to combine intrinsic and extrinsic rewards

## 5.3 AUXILIARY TASKS AND PARAMETER SHARING

As mentioned in Section 3.3, auxiliary tasks can accelerate learning, build a stronger representation, and therefore improve generalization. In UNREAL (Jaderberg et al., 2017), the task is learnt along with several prediction and control tasks such as reward prediction, pixel control and feature control. In Lample & Chaplot (2017), the architecture learns to predict several features of the environment along with the true task and this dramatically improves performance. Note that the auxiliary tasks are only meant to enrich the loss and rarely used to actually predict or control relevant elements. In a different line of work, the UVFA framework (Schaul et al., 2015) introduces the agent's goal as input and effectively shares all network parameters between the different goal-conditioned value functions, which has been used in NGU to range from exploiting to very exploratory policies. Agent57 argued against it, raising the point that decoupling would stabilize learning, allow the function approximators to accommodate their respective reward scales and decouple the optimizer state. Our Shared architecture allows to benefit from the auxiliary task of intrinsic reward learning while addressing the problems raised by Agent57, since each head and associated optimizer can adjust for its own reward scale.

## 5.4 OPTIONS FOR EXPLORATION

Using options for exploration has been probed in prior work, and is often addressed within *option-* or *skill-discovery*, with strong ties to skill learning. The main body of work involves reaching states with key properties. Machado et al. (2017) show that Proto-Value Functions (Mahadevan, 2005) can highlight key states using the MDP Laplacian, and use intrinsic rewards ("*eigenpurposes*") to generate *eigenoptions* to reach said states. Covering Options (Jinnai et al., 2019a;b) use algebraic connectivity to identify the best states to link with options in order to minimize cover time, and build said options with a goal-oriented intrinsic reward. Successor Options (Ramesh et al., 2019) identify *landmark states* that maximize the successor representation, and reach said landmarks using a hill-climbing intrinsic reward. Because they wind down to intrinsic reward learning, all of these methods could be incorporated as Explorers in DeepEOs. From another angle, Nachum et al. (2019) attempt to understand successes of Hierarchical RL, and identify temporally-extended exploration as a potential key element, further motivating our work. Supporting that claim, Dabney et al. (2021) show that simply repeating actions can be a powerful motor for exploration – this could be incorporated in a deterministic Explorer.

## 5.5 COMBINING INTRINSIC AND EXTRINSIC REWARDS

Combining different sources of reward is a problem that has recently gained attention with the rise of more potent IM methods. In Table 1 we analyse several methods previously introduced, using key desirable properties to provide a systematic comparison.

The first column indicates whether the intrinsic and extrinsic Value Functions are separated or not, i.e. using $r = r_e + \beta r_i$ or separate $V_e$ and $V_i$, which can be combined into $V = V_e + \beta V_i$ in a WS setting. As far as we are aware, this idea was first introduced for IM in RND (see 3.1) to combine different discount factors and termination signals. Agent57 later proved that separating the Value Functions stabilizes the training, by allowing the function approximators and optimizer states to accommodate their respective reward scales. This issue was conjointly raised in the original Explore Options paper, proposing to decouple at the policy-level.

The second and third columns show how each method uses the IM signal to explore. `WS` methods perform a weighted sum of the extrinsic and intrinsic signals, which results in a single merged exploring-exploiting behavior. Explore&Exploit (Nachum et al., 2019) was introduced in a vastly different context, to show that Hierarchical RL mainly benefits from exploration. The algorithm also trains an intrinsically motivated Explorer and extrinsically motivated Exploiter and randomly switches the actor every $c_{switch}$ steps. DeepEOs are similar but switch agents using a learnt option.

Column 4 indicates whether the method could transfer the learnt exploratory behavior to a new task. All methods with Combined value functions fail this, since the exploring behavior is engraved into the policy. Since RND separates the value functions at the Critic level, it is unclear whether it could benefit from the learnt $V_i$. Other methods that keep separate value functions can simply extract the learnt Explorer to the new task.

Column 5 indicates whether the method could scale to multiple IM functions $f_{ir}$. `WS` methods fail here, since it would require merging more reward signals and additional layers of $\beta_j$ fine-tuning. Explore&Exploit and DeepEOs naturally scale to multiple IM signals, as it respectively involves switching between more agents, and adding more options.

The last column indicates whether the method shares parameters between the intrinsic and extrinsic reward learners, resulting in an auxiliary task that can improve the representation. We argue that the naive `WS` does not share parameters since it is learning a single value function. Similar to our approach, RND separates the approximator heads, resulting in a shared visual representation. NGU shares all parameters using UVFA. Both Explore&Exploit and Agent57 could benefit from the shared architecture we introduce in DeepEO.

## 5.6 LEARNING FROM MULTIPLE SIGNALS

In Horde (Sutton et al., 2011), the authors propose to build the agent's world knowledge by learning a wide variety of prediction and control tasks that can be useful to learn more complex downstream tasks. This knowledge is represented under the form of value functions, and each learner is referred to as either a control or prediction *demon*. Deep Explore Options are closely linked to the Horde architecture, due to their scalability to several intrinsic reward functions and independent modules. In particular, Explorers can be seen as control demons alongside the Exploiter. Intrinsic reward functions $f_{ir}$ often predict or model elements containing information about the environment, such as the successor representation (Machado et al., 2020), an action-prediction and forward model (Pathak et al., 2017), or density models (Bellemare et al., 2016). All of these can therefore be seen as prediction demons in the Horde architecture. In addition, all of these could be trained using a huge shared representation as auxiliary tasks, and hence building both a world and state representation at the same time. Under this scope, the resulting agent would make direct use of all its demons, providing one solution as to how to use world knowledge – this will be the subject of future work.

## 6 CONCLUSION

Explore Options allow an extrinsically motivated Exploiter agent to call an intrinsically motivated Explorer agent for a set amount of time. They have only been conceptualized in a tabular setting, preventing generalization of the option. We revised them to Deep Explore Options in the Dopamine framework, to learn from intrinsic reward in Deep Reinforcement Learning. We introduced a new multi-agent buffer-selection algorithm, showing that agents can benefit from observing data that is interesting accoding to others. We also introduced a new architecture to combine intrinsic and extrinsic agents, resulting in a $50\%$ faster runtime and stronger representation. To ensure fair comparisons, we followed a benchmarking study, using hard- and easy-exploration games of the Arcade Learning Environment. We showed here that Deep Explore Options are a very strong alternative to the wide-spread practice of a weighted sum of rewards, convincingly beating the baselines in 4 of the 6 tested games, with a good performance on the last two. We also showed that Deep Explore Options can efficiently learn from *several* intrinsic reward signals, ignore harmful signals, and extract their exploiting behavior; all while maintaining a strong shared representation. We compared our introduced method to existing work in intrinsic motivation and discussed desirable properties of methods to combine extrinsic and intrinsic signals.

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

## A  IMPLEMENTATION DETAILS

### A.1  MINOR ALGORITHMIC CHANGES

#### A.1.1  FIXING $\epsilon$-GREEDY ACTION-SELECTION

$\epsilon$-greedy is one of the most naive exploration methods, where we simply uniformly try out a random primitive action with probability $\epsilon$. Since our option persists over $c_{switch}$ steps, while other actions only last 1 step, this needs to be balanced for. We provide a more complete and precise understanding of the $p_{ex}$ hyperparameter provided in the original Explore Options paper, which allows for a more logical choice of its value.

What is desirable is not uniform action selection, but uniform *agent interaction* when exploring with $\epsilon$-greedy, since towards the earlier stages of learning, we do not know which agent will best find rewards. We can achieve this simply by tweaking the option probability $p_{ex}$ (defaulting at $1/(|A|+1)$). When $N$ steps are performed randomly by the Exploiter, an expected $N p_{ex} c_{switch}$ are performed by the Explorer. To achieve a ratio of $r = 50\%$, we therefore need $r = p_{ex} c_{switch} / (1 + p_{ex} c_{switch})$; replacing $p_{ex} = \rho / c_{switch}$ gives $\rho = r / (1 - r)$. We use a linear schedule of $r$ from 0.5 to 0.001 over 200 million steps to ensure that the call to the Explorer is frequent at first, but eventually up to the Exploiter.

#### A.1.2  $n$-STEP AND OPTION LEARNING

Rainbow uses $n$-step learning, which bootstraps $Q$ value estimates after $n$ steps of real-world transition data. This is well-known to achieve a balance between bias and variance for the right value of $n$. In our case, this means that we need to learn from sequences of $n$ transitions in which the option can appear among primitive actions (e.g. $a \rightarrow o \rightarrow a$ or $a \rightarrow a \rightarrow o$ at $n = 3$). We empirically find that the agent does *not* benefit from such $n$-step learning of the option. We infer this is because it adds dramatic variance to all primitive actions, as the option's value is hard to predict. Therefore, the $n$ transitions before bootstrapping do not contain the option and the option is only learnt using one-step transitions. So in practice, when the Exploiter uses the Explore Option, the Explorer acts for $c_{switch}$ time-steps, then gives the control back to the Exploiter. This option-transition $(s, a, r, s') = (s_t, o, R, s_{t+c_{switch}})$ with $R = \sum_{k=0}^{c_{switch}-1} \gamma^k R_{t+k+1}$ is added to the buffer like other transitions, simply adjusting the discount during bootstrapping.

### A.2  HYPERPARAMETERS & ARCHITECTURES

| Hyperparameter | Value |
|---|---|
| Discount factor $\gamma$ | 0.99 |
| Min history to start learning | 80K frames |
| Target network update period | 32K frames |
| Adam learning rate | $6.25^{-5}$ |
| Adam $\epsilon$ | $1.5^{-4}$ |
| Multi-step returns $n$ | 3 |
| Distributional atoms | 51 |
| Distributional min/max values | $[-10, 10]$ |
| $\epsilon$-greedy schedule | $1 \rightarrow 0.01$ over 1M frames |

Table 2: List of Rainbow hyperparameters used, taken from the `benchmark`

The list of used Rainbow hyperparameters can be seen in Table 2 above. The network architecture used for the `Separate` configuration is the Categorical DQN, as used in Dopamine (Rainbow without Dueling). The network has 3 Convolutional layers followed by 1 fully-connected layer, which we refer to as the "vision module", followed by a fully-connected output layer, which we call "control module". The Exploiter has one more action per Explorer. Instead of keeping a separate network for Exploiter and Explorers, the `Shared` and `Multi` architectures simply share the vision module between all agents and branch out a control module for each agent.

# B   ADDITIONAL RESULTS & VISUALS

## B.1   EXPLORER & OPTION INSIGHTS

The Explorers are also implemented as Rainbow agents using their respective intrinsic rewards as learning signals, and without a `done` life-termination signal. When their respective options are chosen by the Exploiter, they are called to act for $c_{switch}$ steps. We can therefore track how frequently they act in the environment, providing insight as to how relevant each Explorer is to the Exploiter. We can also keep track of the Explorer's Q-values during training (on training batches), giving an idea of how interested each agent is in the environment at each step of training. We compute both metrics for each Explorer every million steps.

The visuals for the `Multi` architecture and for all games are provided in Figure 3, with Explorer1 (left column of each game) referring to the RND-based Explorer, and Explorer2 (right) referring to the `ConstNeg`-based Explorer.

There are several key observations to be made – first, we can infer how interesting a game was to the Explorers based on their Q-value estimates (top row of each game). The `ConstNeg`-based Explorer in particular allows us a degree of interpretability: if the Q-value estimate skims the $-2.5$ mark, it means the Explorer is not interested at all in the environment and manages to correctly predict its reward function. However, in certain games, especially hard-exploration ones (PRIVATEEYE, MONTEZUMA, GRAVITAR), we see a tendency to overestimate the value of the observed states. We understand that this is because these games display new visuals on rare occasions, which makes these parts of the input space hard to predict. Interestingly, the RND-based Explorer seems to align with the `ConstNeg`-based Explorer, as it has a more noisy and generally higher Q-value estimate in the same games.

To a certain extent, we can also infer how relevant the Exploiter finds each Explorer based on the interaction frequencies, i.e. how many times the Explorer acted in the environment over a period (bottom rows). These vary wildly, between near-zero in some games to up to $20\%$ in others. This, however, more likely provides an idea of how indulgent the environment is, as very hard environments wouldn't allow for a non-exploiting policy to be tolerable (leading to death or huge performance losses). So this measures how much the agent can afford to explore without big losses of performance. A better measure of the relevance of the Explorers to the Exploiter is the direct ratio of performance of the Exploiter with and without Explorers, as reported in the main text in Figure 2. Combined with the observations of Figure 3 here, we can see that hard-exploration games tend to be also hard to navigate, since they allow for little leeway for Explorer interactions, but they absolutely require the Explorers, since the Exploiter performance crumbles without them. In easy-exploration games or GRAVITAR (which is more arguably hard-exploration), the Exploiter can afford to use the Explorers much more, but doesn't require them as critically, in fact performing much better in isolation. Still, Explorer interactions of more than $5\%$ show that the Exploiter finds a clear utility to using the Explorers.

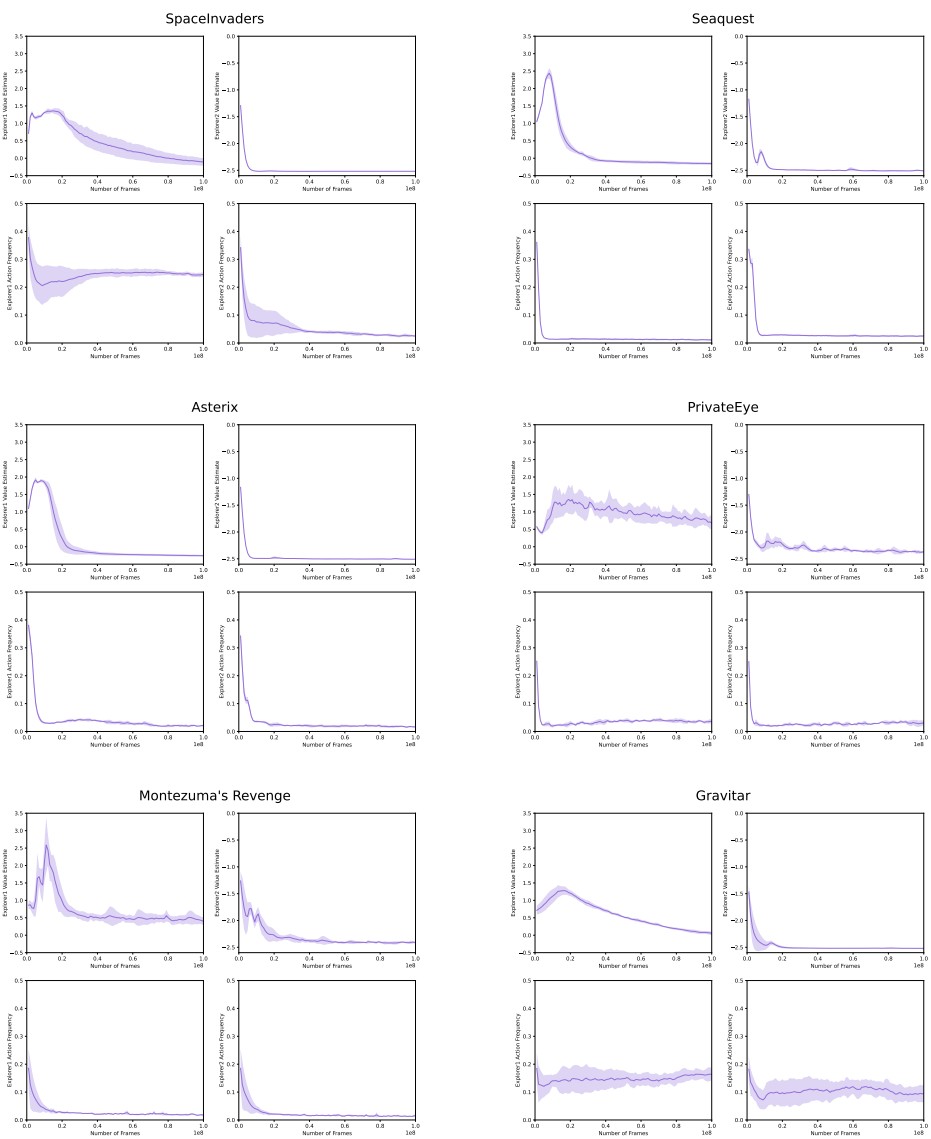

Figure 3: Explorer interaction frequency and Q-value estimates in all games for the `Multi` architecture.

