# OpenReview forum: "Deep Learning of Intrinsically Motivated Options in the Arcade Learning Environment"
_ICLR.cc/2022/Conference — ICLR 2022 Submitted_

### Official Review · Reviewer_jgim · 2021-10-27

**Correctness:** 2
**Technical Novelty And Significance:** 2
**Empirical Novelty And Significance:** 2
**Recommendation:** 3
**Confidence:** 4

**Main Review:**

This paper introduces a new agent architecture that has several moving pieces and it doesn’t do a good job at evaluating their impact. In the end it is hard to claim that the value of the main idea (to use an intrinsic reward signal without having to linearly combine it to the extrinsic reward) was backed up with clear experiments. There are several confounders. For example, it could still be that the observed empirical performance is due to some sort of temporally-extended exploration, and that’s all. I don’t see data to rule this out. Finally, the paper makes several claims that are not precise or that are not supported by data, including some of its claimed contributions. I elaborate on these topics below.

First and foremost, I don’t think we have data to quantify the impact of several components discussed in the paper. Dabney et al. (2021) has shown the benefits of repeating the same action multiple times to generate temporally-extended exploration. When I visit their paper, and I look at their learning curves (Fig. 19 from this version https://openreview.net/pdf?id=ONBPHFZ7zG4), I noticed that after 100M frames, also using Rainbow, they report a better or similar performance to DeepEO in 4 out of 6 games (Asterix, Gravitar, Seaquest, and Space Invaders). So, could it be that everything that is being reported here is nothing but a consequence of temporally-extended exploration induced by options? This is obviously the most glaring concern, but there are others. For example, J-PER is cast as a contribution but it is never actually evaluated as such, I believe. The least I would expect is a comparison between J-PER and uniform sampling. Finally, on this topic, I don’t think the impact of ConstNeg was properly evaluated as well. I mean, one could see a uniform penalty throughout the state space as an optimistic initialization, which drives exploration (c.f. Domain-Independent Optimistic Initialization for Reinforcement Learning by Machado, Srinivasan, and Bowling). It is not necessarily bad. I wish there was some study about the impact of such a signal (or the generated option). Just casting it as bad to contrast it to a regular weighting of intrinsic rewards is dangerous.

In terms of overclaiming, I struggle to find how using multiple heads to predict the value of different cumulants (to use GVF nomenclature) can be seen as a contribution. This is a fairly standard approach to deep RL thus, is it really a contribution? Based on my previous comment on J-PER and this one, I believe the last two contributions stated in the paper are not justified. Maybe the first one is not as well, given the ez-greedy results.

For the method itself, it seems that both c_switch and w_s have a big impact on performance (Fig. 2). It is said that these parameters are easy to tune, but how can that be if there’s such variability across values close to each other? It is particularly tricky to balance loss functions, mainly when multiple intrinsic reward signals are considered. It seems to me that the proposed method has several parameters that might make it brittle. Importantly, operating at different time scales seem to be an important ability that we should not prevent our agents to have with a fixed c_switch value.

Finally, besides what I already pointed out in the previous paragraph, the write-up itself is sometimes imprecise and some of the statements are not backed up. To name a few:
- “explicitly looking for new knowledge and experiences”: I don’t know what knowledge is in this context, it is never defined. What is the difference to experience, for example?
- “such as wasting world knowledge and transfer potential”: Is world knowledge different than “knowledge”? What is “transfer potential”? Was this evaluated in any experiment to justify this claim?
- [Multi] “builds a stronger representation”. Is there any experiment to back this up? What is even a good representation?

**Summary Of The Paper:**

This paper introduces an extension of Explore Options (EO; Bagot et al. 2020) to the non-linear function approximation case. EO consists in training two different policies in parallel, one to maximize the intrinsic reward, and one to maximize the environment's reward. Then, the agent can decide to act according to the option generated by the intrinsic reward for a fixed number of time steps, which provides it with temporally-extended exploration and bypasses the need to combine different reward signals by tuning a scalar value.

The method proposed in this paper extends EO by designing a neural network that has a shared torso and J different heads, one for each value being learned (environment's reward and intrinsic reward, for example). To do so, because it wants to use PER, it needs to adapt PER to deal with errors from multiple heads. It does so by keeping track of the errors in the two different heads and sampling transitions according to the J error values to ensure samples interesting for each of the J heads are present in the minibatch.

Empirical results show that the proposed method tend to outperform standard RND. This is relevant because the intrinsic reward function used to generate the exploratory option was the RND loss so, in a sense, this is a direct comparison between linearly combining reward signals and using them as an option.

**Summary Of The Review:**

This paper introduces a new agent architecture that has several moving pieces and it doesn’t do a good job at evaluating their impact. In the end it is hard to claim that the value of the main idea (to use an intrinsic reward signal without having to linearly combine it to the extrinsic reward) was backed up with clear experiments. There are several confounders. For example, it could still be that the observed empirical performance is due to some sort of temporally-extended exploration, and that’s all. I don’t see data to rule this out. Finally, the paper makes several claims that are not precise or that are not supported by data, including some of its claimed contributions.

---

> ### Author Response · Authors · 2021-11-24
> **Thank you for the helpful feedback**
>
> Thank you very much for reading through our paper and your feedback. The point raised in your review are extremely relevant - we would like to address some of them, to build a stronger version of the draft next time.
>
> The comments on over-claiming are absolutely fair and we take them to heart for our next version.
>
> - regarding the actual contribution of an intrinsically motivated agent versus simple temporally-extended exploration, would it help to have videos or images showing the independent behavior of the intrinsically motivated agent? This might also help in making the Transfer claims clearer.
> - J-PER was indeed not evaluated properly and we will correct this in later versions
> - feedback on the ConstNeg reward is very interesting - we indeed should not have framed it as generally dangerous, and kept clear that it was only so for in a weighted sum of rewards.
> - in the intrinsic motivation literature, due to the dominance of the weighted sum of rewards, it is very rare to have several cumulant heads and we therefore used this as a basis to claim novelty; we were unaware that it was a standard approach in other sub-fields. This will be adjusted in later versions.
> - the worries in the stability of the method to the introduced hyperparameters is very justified and our next drafts will be clearer and fix this.
> - operating at different time-scales is of interest, but the focus of our work was not on Hierarchical RL - the option is meant to provide data to learn off-policy, rather than be part of the final optimal policy. As stated above, including options was rather to benefit from directed, temporally-extended exploration.
> - the comment on imprecise writing is also taken to heart and will be a major axis of improvement. We were basing our statements on the Horde and Auxiliary Task learning literature, that often mention "building up world knowledge" and a "stronger, more general" representation, but we understand that these are hand-wavy and should be properly defined and backed
>
> We again thank you for this excellent review and will strive to take it all into account in subsequent versions of the work.

---

> ### Comment · Reviewer_jgim · 2021-11-29
> **Final Assessment**
>
> I've gone over the authors' response and I'm keeping my final score/recommendation for this paper. The authors agree the paper is not ready for publication.

---

### Official Review · Reviewer_yuhD · 2021-10-31

**Correctness:** 3
**Technical Novelty And Significance:** 2
**Empirical Novelty And Significance:** 2
**Recommendation:** 3
**Confidence:** 2

**Main Review:**

1.

The explore options (EO) or deep EO generalization done in this paper seems closely related to semi-MDPs. There is no reference or contextualization w.r.t this literature.

Why is a new formalism needed if it can already be formalized as an semi-MDP? If semi-MDPs are not rich enough to describe these ideas then what is missing?

2.

Figure 2 provides quantitative results and I was surprised to see that the scores on montezuma's revenge are not as high.

There have been other approaches (e.g. Go-explore) that has surpassed this by a wide margin or other hierarchical RL approaches (e.g. h-DQN) that have studied ways to explore more sample efficiently.

On a related note, I think the biggest limitation of this paper is that the experimental validation is not thorough. Where were these particular 6 games selected? It is difficult to quantify the merits of this approach without running this on wider environments.

3.

Since the key idea is to learn options, it would have been interesting and important to visually inspect the type of intrinsic options that are learned by this approach. The authors provide a partial answer via Figure 3 in the appendix but this does not clearly indicate the exploration abstract learned by this approach

4.

Is the exploration option's termination condition always deterministic? This seems like a big limitation in both discrete and continuous domains. What are the effects of changing this parameter on the performance beyond Gravitar?

**Summary Of The Paper:**

This paper proposes a directed exploration algorithm within the context of Deep RL. It learns two separate off-policy agents -- one learning to explore and the other one to exploit given task-specific rewards. The explorer is task-agnostic and assumes that a set of intrinsic rewards are provided. The exploiter can execute raw actions or options parametrized by the explorer. The approach is shown on a small set of atari games, where the key performance metric is the highest score reached within 1e8 number of frames.



**Summary Of The Review:**

I think the ideas presented in this paper are interesting and promising. However, I would like to see them validated on more than 6 ALE games -- currently the results are somewhat mixed (helps on 4/6 to some extend) and there isn't an empirical result that very clearly demonstrates a new capability enabled by this approach.

The paper already needs to be more clearer written and framed w.r.t a rich literature on semi-MDPs and hierarchical RL.

---

> ### Author Response · Authors · 2021-11-24
> **Thank you for the helpful feedback**
>
> Thank you very much for reading through our paper and your feedback. We would like to address some of the points raised, to build a stronger version of the draft next time.
>
> 1. Explore Options are absolutely part of the options (semi-MDP) framework, we did not mean to introduce them as something separate, and we did mention the Option framework in the background and related work sections, though we understand we should have been more thorough. The original tabular paper did go into a bit more depth regarding their instantiation of options - since their construction is relatively straight-forward, it was not included in this draft.
>
> 2. the 6 games selected were half of the 12 mainly studied games from the Benchmark paper, selecting 3 hard-exploration and 3 easy-exploration games were the baseline methods showed most variance between each other, and therefore where work most likely had to be done. We selected only 6 since Atari is an extremely expensive environment and we could not afford more; no results on other games were computed or they would have been published along.
> We believe the comparison with Go-Explore is a bit unfair as our focus was on comparing a weighted sum of rewards with option-learning, in the Benchmark paper's experimental setup of comparing intrinsic motivation methods.
>
> 3. We will include visualization of the learnt behaviors in subsequent drafts
>
> 4. We understand that broader hyperparameter studies would be desirable, and we will try to compile as much as our resources can allow. c_switch is indeed always deterministic, but we believe that more complex termination functions are outside the scope of this submission since making the method more complex would reduce its practical pertinence and might confuse the straight-forward nature of our claim
>
> We thank you again very sincerely for your helpful review and will take your feedback into account for subsequent versions of this work.

---

### Official Review · Reviewer_zfem · 2021-11-02

**Correctness:** 2
**Technical Novelty And Significance:** 2
**Empirical Novelty And Significance:** 3
**Recommendation:** 5
**Confidence:** 4

**Main Review:**

Strengths:
- Important problem, definitely of interest to the ICLR community
- Care seems to be taken to follow best practices with regards to conducting RL experiments (random seeds & using standard implementations)

Weaknesses:
- Limited performance improvement in the settings where the greatest benefit would be expected (hard exploration problems)
- Limited technical novelty
- Experiments could be more comprehensive (only compared with 2 baselines, ablations are relatively limited)
- Some claims (e.g., "improved representation" and instability of weighting multiple IM objectives) are not clearly backed up by experiments

Other comments:
- "We combine the losses through L = Le + λLi with λ = 1." The loss terms here are not defined
- "This, again, is not to be confused with the β from WS, since each head adjusts for their own reward scales, meaning that λ does not carry the duty of reward balancing." This is still a sort of balancing, because if the rewards for explorer are much larger/smaller than the exploiter, the gradients contributions will also be much larger/smaller, right?
- "All observed transitions are added to the same buffer, regardless of the agent currently interacting." Which reward is stored in the buffer? Or are both rewards stored?
- "where the tuning of all βj and the cacophony of all reward signals would make finding a stable agent virtually impossible" It would be nice to back up this claim with experimental evidence
- It's hard to interpret the results of Figure 2- on one hand, max(Exploiter only, DeepEO Multi) seems to be fairly strong; however, I'm not sure this is the best interpretation. It seems problematic that in some cases, the exploitation policy is much stronger than the explore option policy, while in others, relying on the exploration policy is necessary to achieve good performance. In practice, which should be used when the policy is deployed?
- The algorithm introduces 3 additional hyperparameters, \lambda, c_{switch}, and w_s. They don't seem to require significant tuning, but given that the algorithm is partially motivated by the difficulties incurred by the weighting parameter in standard intrinsic motivation methods, it seems worth mentioning as a minor downside.
- "Our Shared architecture allows to benefit from the auxiliary task of intrinsic reward learning while addressing the problems raised by Agent57, since each head and associated optimizer can adjust for its own reward scale." I'm unconvinced this is true- even with separate heads, since the majority of the network is shared, aren't different reward scales a problem?
- "...resulting in an auxiliary task that can improve the representation." This claim doesn't really seem to be clearly supported by the experiments, unless I'm missing something
- The primary benefits of "resulting in a 50% faster runtime and stronger representation" don't seem to be systematically evaluated in the experiments.
- "...convincingly beating the baselines in 4 of the 6 tested games, with a good performance on the last two." It's counterintuitive that the main benefit from DeepEO came on the *easy* exploration tasks, rather than the hard ones. Can the authors explain why this occurs? It seems counter to the idea that separating IM is important for exploration.

**Summary Of The Paper:**

The paper is motivated by the challenging problem of exploration in reinforcement learning. The authors build on previous work in learning exploration options, extending it to the setting of function approximation and learning from image-based observations. The primary contributions of the work are the function-approximation version of Explore Options, called Deep Explore Options (DeepEO), which learns a separate exploration policy alongside a pure exploitation policy. In addition, the authors describe the training strategies/data sampling mechanism needed to train DeepEO successfully.

**Summary Of The Review:**

Overall, the authors attack an important problem of interest to the ICLR community. While the idea of using options for better exploration in deep Q networks is interesting, the proposed method does not currently provide clear performance improvements over existing approaches in the experimental settings shown, and the experimental settings themselves were relatively limited in terms of the diversity of baselines considered. In addition, some claims made in the paper are not fully backed by experimental evidence. Thus, I recommend the current version of the paper be rejected.

---

> ### Author Response · Authors · 2021-11-24
> **Thank you for the helpful feedback**
>
> Thank you very much for reading through our paper and your feedback. We would like to address some of the points raised, to build a stronger version of the draft next time.
>
> First and mainly, we take your feedback on unbacked claims at heart and this will be one of our main points of attack for a subsequent version of the paper. We thank you again deeply for this insight. The "stronger representation" was mainly taken from Auxiliary Task Learning, where the learning of several tasks ensures generalization to an extent, but this is indeed something that should have come with empirical backing.
> Addressing some of the most crucial points raised:
> - While it is true that combining losses contains some of the balancing issues of the weighted sum of rewards, we rely on empirical evidence from the Auxiliary Task learning community to claim that dozens of completely unrelated losses can still work together to build a joint representation.
> - Both (or all) rewards are stored in the buffer, as indicated by Fig 1; this is crucial for off-policy learning of the Exploiter and Explorers
> - It is true that the fact that there isn't a single agent that consistently performs better of Exploiter-only and Multi can be a hindrance in production, however if this is the only switch to figure out, we believe it is a major step from fine-tuning exploration parameters with a large  and sensitive spectrum like β
> - mentioning the inclusion of new hyperparameters as a downside is indeed necessary and will be done in future drafts
> - the question of lower performance on hard-exploration games is still partly open to us, but we suspect that near-offline learning of the Explorer might be one of the culprits. However, Montezuma is a separate case, since as the Benchmark paper indicated, it has been used as its own benchmark for hyperparameter tuning in most of the IM methods, so it is understandable that our general method wouldn't compare to heavily-finetuned algorithms.
>
> We thank you again for this most helpful feedback, and will take it into account for subsequent drafts of the paper.

---

> ### Comment · Reviewer_zfem · 2021-11-29
> **Final comment**
>
> I appreciate the authors' clarifications and receptiveness to the feedback provided in the reviews. I hope future versions of the work will benefit from the comments from the review period. I will keep my score.

---

### Official Review · Reviewer_svsx · 2021-11-04

**Correctness:** 2
**Technical Novelty And Significance:** 2
**Empirical Novelty And Significance:** 2
**Recommendation:** 3
**Confidence:** 3

**Main Review:**

Strengths
--
- The main motivation of creating a method to incorporate multiple intrinsic reward signals is good
- The proposed method makes intuitive sense -- simply learn separate policies from each reward signal off-policy
- The proposed 2-PER method exhibits a performance improvement in the environment it was tested in

Weaknesses
--
- W1 A variety of the claims are unsupported due to missing evidence or ambiguity in the experiments
- W2 There are several important writing issues

Weakness 1
--
- W1.1 It's odd to claim the method is J-PER, yet only apply it to the 2-PER setting. It's also not clear how to implement the proposed method beyond J=2 setting.
- W1.2 The final set of intrinsic rewards used in the main method is unclear. Is it just RND and ConstNeg? Because of this ambiguity, it's also unclear whether the method's performance improves with the addition of more intrinsic rewards (a claimed contribution).
- W1.3 The comparison is weak -- only two baseline exploration-bonussed methods are compared to. It would be good to include a skill-learning method. For instance [A] can be considered a method for learning intrinsically motivated options and then planning over them to achieve extrinsic rewards.
- W1.4 The differences between the architectures (Shared, Separate, and Multi) are unclear. The main paper needs to be clearer about this point.
- W1.5 The sweep of $w_s$ is only performed in one environment. This makes it hard to draw the conclusion that the proposed 2-PER algorithm is generally useful.

Weakness 2
--
- W2.1 The related work section is 2 pages long, which is seems too long for a 9-page paper. I think its size could be significantly reduced.
- W2.2 It's unclear why copying the buffer once for each agent is an important bottleneck.
- W2.3 These losses ($\mathcal L_e$ and $\mathcal L_i$) have not yet been defined.
- W2.4 The writing is generally informal and meandering. I recommend trying to make the writing much more concise. Otherwise, readers with less patience are likelier to ignore this work.

Minor weaknesses
--
- Top of page 5: the use of an ellipsis here is too informal.

[A] Sharma, Archit, et al. "Dynamics-aware unsupervised discovery of skills." arXiv preprint arXiv:1907.01657 (2019).

**Summary Of The Paper:**

This paper proposes an approach to learning intrinsically motivated options that is combined with an extrinsically trained policy. The motivation is to formulate a learning algorithm that can scale to the availability of multiple intrinsic reward signals better than an approach that attempts policy learning with a linear combination of extrinsic and intrinsic rewards. Experiments on several ALE environments illustrate that the method can perform better than a baseline method that incorporates an intrinsic reward with a weighted sum.

**Summary Of The Review:**

While the motivation and method makes sense, many of the claims are unsubstantiated, weakly substantiated, or ambiguously substantiated (see Weakness 1). Because the focus of the paper is mainly on demonstrating the empirical utility of the proposed method, significant improvements are needed to make the paper acceptable.

---

> ### Author Response · Authors · 2021-11-24
> **Thank you for the useful feedback**
>
> Thank you very much for reading through our paper and your feedback. We would like to address some of the points raised, to build a stronger version of the draft next time.
>
> W1.1. We did use 3-PER for the "Multi" agent using 2 intrinsically-motivated Explorer; our apologies if this was not clear enough. Clarifications regarding J>2 were removed to save space, but should have been shifted to the Appendix. We will take all of this into account in a future draft
> W1.2. and W.1.5. we understand the need for more games, intrinsic motivation functions and broader hyperparameter studies; we would like to note that Atari games are tremendously expensive environments and such additional runs would freeze research resources for months.
> W1.3. Our point in this paper was to take down the usual method of a weighted sum of rewards when introducing a new intrinsic rewards, and to motivate usage of several independent rewards. We believe general skill-discovery methods are outside our scope. We also tried our best to work on top of the Benchmark paper, to use a well-founded experimental setup as basis; the mentioned method doesn't appear in the benchmark and would therefore require a potentially dubious reproduction in the Dopamine framework.
> W1.4 W2.4 we thank you very much for comments on the quality of the writing and will take them into consideration swiftly.
> W2.1 Another reviewer was asking for more details in one of the sections of the Related Work; we will try to find a balance. We do understand the section is very long, but the kind of higher-level study of desirable properties of intrinsic motivation we make wasn't found in previous work.
> W2.2 We meant to say that PER sampling is an expensive method, as it samples the transitions one by one. Repeating the sampling of a full batch for a big number of agents linearly scales with that amount of agents. This explanation will be made clearer in future versions.
> W2.3 Li and Le were included as placeholders for whatever intrinsic and extrinsic losses we want to use, in a similar fashion to Ri and Re for intrinsic and extrinsic rewards; this will also be made clearer
>
> We thank you again for your very helpful and thorough review; we will make sure to incorporate answers to our weaknesses in subsequent versions of our paper.

---

### Decision · Program_Chairs · 2022-01-20

**Decision:**

Reject

**Comment:**

This work gives an interesting perspective on combining options with exploration in the non-tabular case. The reviewers have raised a number of important areas for improvement (primarily missing ablations to support the claims of the paper, but also specific suggestions about improvements to the text), and feel that sufficient work is required to address these that the paper should be rejected at this time.